



# Prospects for dendroanatomy in paleoclimatology – a case study on *Picea*
*engelmannii* from the Canadian Rockies
Kristina Seftigen[1,2*], Marina V. Fonti [2,3], Brian Luckman[4], Miloš Rydval[5], Petter Stridbeck[1], Georg von Arx[2,6], Rob Wilson[7], Jesper
Björklund[2]
[1] Regional Climate Group, Department of Earth Sciences, University of Gothenburg, Gothenburg, Sweden.
[2] Dendrosciences, Swiss Federal Institute for Forest Snow and Landscape Research WSL, Switzerland
[3]Institute of Ecology and Geography, Siberian Federal University, Krasnoyarsk, Russian Federation
[4] Department of Geography, University of Western Ontario, London, ON, N6A 3K7, Canada
[5] Department of Forest Ecology, Faculty of Forestry and Wood Sciences, Czech University of Life Sciences Prague, Prague,
Czech Republic
[6]Oeschger Centre for Climate Change Research, University of Bern, Switzerland
[7]University of St Andrews, North Street, St Andrews, KY16 9AL, UK
*Corresponding author:
E-mail address: kristina.seftigen@gvc.gu.se
**Abstract**
The continuous development of new proxies as well as a refinement of existing tools are key
to advances in paleoclimate research and improvements in the accuracy of existing climate
reconstructions. Herein, we build on recent methodological progress in dendroanatomy – the
analyses of wood anatomical parameters in dated tree rings – and introduce the longest (1585
– 2014 CE) dendroanatomical dataset currently developed for North America. We explore the
potential of dendroanatomy of high-elevation Engelmann spruce (*Picea engelmannii*) as a
proxy of past temperatures by measuring anatomical cell dimensions of 15 living trees from
the Columbia Icefield area. There, X-ray maximum latewood density (MXD) and its blue
intensity counterpart (MXBI) have previously been measured, which allows comparing the
different parameters. Our findings highlight anatomical MXD and maximum radial cell wall
thickness as the two most promising wood anatomical proxy parameters for past
temperatures, each explaining 46% and 49%, respectively, of instrumental, high-pass filtered,
July-August maximum temperatures over the 1901-1994 period. While both parameters
display comparable climatic imprinting at higher frequencies to X-ray derived MXD, the
anatomical dataset distinguishes itself from its predecessors by providing the most temporally
stable warm-season temperature signal. For the long-term secular trends, discrepancies
between anatomical MXD and maximum radial cell wall thickness chronologies were
observed, where the former more closely follow the long-term variations of the X-ray based
MXD. Further studies, including samples from more diverse age cohorts and the adaptation
of RCS-based standardizations, are needed to disentangle the ontogenetic and climatic
components of long-term signals stored in the wood anatomical traits and to more
comprehensively evaluate the potential contribution of this new dataset to paleoclimate
research.



**Keywords:** Dendroanatomy, *Picea engelmannii*, Canadian Rockies, tree rings, latewood density, temperature reconstruction, paleoclimatology

## 1. Introduction

Tree rings form the backbone of high-resolution palaeoclimatology of the Common Era by providing precisely dated, annually resolved, spatially widespread and easily accessible archives of climate proxy data. Tree-ring archives make up more than half of all publicly available temperature proxy records and are greatly influential in multi-proxy hemispheric-scale temperature reconstructions (PAGES 2k Consortium 2017). They are vital for spatially explicit mapping of important climate periods (e.g., PAGES 2k Consortium 2013), and the study of temporally distinct cooling events caused by volcanic eruptions (e.g., Schneider et al. 2015; Stoffel et al. 2015; Wilson et al. 2016). Moreover, tree-ring based climate reconstructions play a key role in many of the emerging proxy-model comparison efforts (e.g., Goosse 2017; Luterbacher et al. 2016; Pages k-PMIP3 group 2015; Phipps et al. 2013; Seftigen et al. 2017).

The most frequently and successfully used tree-ring parameters for the study of temperature variations at high latitudes and altitudes are ring width and maximum latewood density or simply maximum density (MXD) (e.g., Esper et al. 2018). While ring width is the most easily acquired measure of year-to-year variations in climate, the parameter often proves difficult to interpret as it may represent distorted transformations of the underlying climate (e.g., Frank et al. 2010; Lücke et al. 2019). In particular, ring width may exhibit amplified low-frequency signals (von Storch et al. 2004) resulting from lagged growth processes in response to climate (Esper et al. 2015) or non-climatic processes (Rydval et al. 2015). Consequently, the presence of prominent decadal variability should not be taken as evidence of corresponding variability distribution in climate observations, and an overestimation of low-frequency signals is often observed (e.g., Franke et al. 2013; Seftigen et al. 2017; Wilson et al. 2016). The MXD parameter, in contrast, generally contains a stronger climate signal with higher signal-to-noise ratios (e.g., Briffa et al. 2002; Ljungqvist et al. 2020), as well as less biological persistence (Esper et al. 2015) and age-related signal-muting (Konter et al. 2016), and is less influenced by stand disturbances (Rydval et al. 2018). However, a number of recent studies (Björklund et al. 2019) (Edwards et al., 2021, in review) have proposed the accuracy of the MXD parameter to be sensitive to measurement resolution. Björklund et al. (2019) showed that increasingly lower resolution of MXD data could result in an increased artificial similarity to the climate response of ring width, and thus that several of the issues facing ring width as a climate proxy may also represent non-negligible constraints on the MXD parameter.





To reduce uncertainties, future reconstruction efforts could profit from the development of new
proxy types and parameters for paleoclimatology, as well as new and expanding
methodologies. Recently, dendroanatomy – the analyses of wood anatomical traits in dated
tree rings (Fonti et al. 2010; Pacheco et al. 2018) – have become more accessible through
semi-automated approaches to quantify wood cell anatomy (Prendin et al. 2017; von Arx and
Carrer 2014; von Arx et al. 2016). Analysis of anatomical cell dimensions is now possible at
the scale required for high-quality climate reconstructions over centuries to millennia
(Björklund et al. 2020). Unlike ring width, anatomical traits of temperature-limited conifers
appear to be less affected by biological memory effects and are imprinted with strong and
mechanistically-grounded temperature signals (Björklund et al. 2019; Cuny et al. 2019; Cuny
et al. 2014). Moreover, cell anatomical measurements have unprecedentedly high temporal
resolution relying on the base unit of the xylem – the tracheid cell, and their biological
foundations and functional links are comparably well understood (e.g., Bouche et al. 2014;
Pittermann et al. 2011; Wilkinson et al. 2015).

In this article, we aim to explore the value of dendroanatomy for high-elevation living
Engelmann spruce (*Picea engelmannii)* trees as a proxy of past temperatures. We make use
of tree samples from the Columbia Icefield area of the Canadian Rockies (Fig. 1) – a site
known for hosting the longest (950-1994 CE) available temperature-sensitive tree-ring
densitometric collections for boreal North America (Luckman et al. 1997; Luckman and Wilson
2005). The Icefield collection, originally comprising ring width and MXD measurements, have
previously been used in regional (George and Luckman 2001; Luckman 1997; Luckman 2000)
and hemispheric-scale (Briffa et al. 2002; D'Arrigo et al. 2006; Esper et al. 2002; Mann et al.
1999) temperature reconstructions. It has additionally been included as one of the key proxy
sites in recent large-scale Northern Hemisphere summer temperature reconstruction
syntheses (Anchukaitis et al. 2017; Schneider et al. 2015; Wilson et al. 2016). The analysis of
the new dendroanatomical dataset produced here includes an assessment of its signal
strength and the imprint of temperature within a number of wood anatomical traits in a well
replicated (N = 15 trees) dataset with dendroanatomical standards, spanning the period 1585
– 2014 CE. We detail common variance amongst selected anatomical parameters, and
emphasize the reconstruction potential of this dataset. The availability of MXD from the
Columbia Icefield area (Luckman et al. 1997; Luckman and Wilson 2005) produced with the
state-of-the-art Walesch Electronic Dendro2003 technique (Eschbach et al. 1995) and its
predecessor (Schweingruber et al. 1978) (hereafter referred to as X-ray MXD), and latewood
blue-intensity (referred to as MXBI) (McCarroll et al. 2002) measurements allow here for an
optimal opportunity for testing the skill and potential advantages of dendroanatomical



parameters as climate proxies. This work is part of a larger ongoing collaborative effort
dedicated to developing a network of long (~500-1000 years) wood dendroanatomical
chronologies from a number of pivotal locations across the northern hemisphere. The ultimate
ambition of this initiative is to sharpen signal interpretations of the dendrochronological records
and optimizing seasonal and temporal fidelity of the proxy-based reconstructions in order
revise (or reinforce) previous conclusions about pre-industrial climate variability and the
mechanisms causing this variability. This work also represents a first step towards a
millennium long anatomical *P. engelmannii* dataset for the Columbia Icefield area, Canada.

## 2. Data and methods

### 2.1 Sample preparation and dendroanatomical measurements

Fifteen living *P. engelmannii* trees (one core per tree) were selected for dendroanatomical
measurements from a collection sampled in 2015, from tree-line sites (2000–2100 m a.s.l.)
adjacent to the Athabasca Glacier in the Columbia Icefield area of the Canadian Rockies
(52.13 N, 117.14 W) (Fig. 1). The selection of cores was based on 1) the visual appearance
of the material (cores with obvious defects were avoided), 2) the temporal coverage of the
series (we strived to have an even replication through time) and, 3) the common signal
strength based on RBAR statistics (Wigley et al. 1984) of the ring-width measurements (in
general, cores with higher than average RBARs were selected for wood anatomy). The
selection was primarily dictated by 1) and 2), and only secondarily by 3).

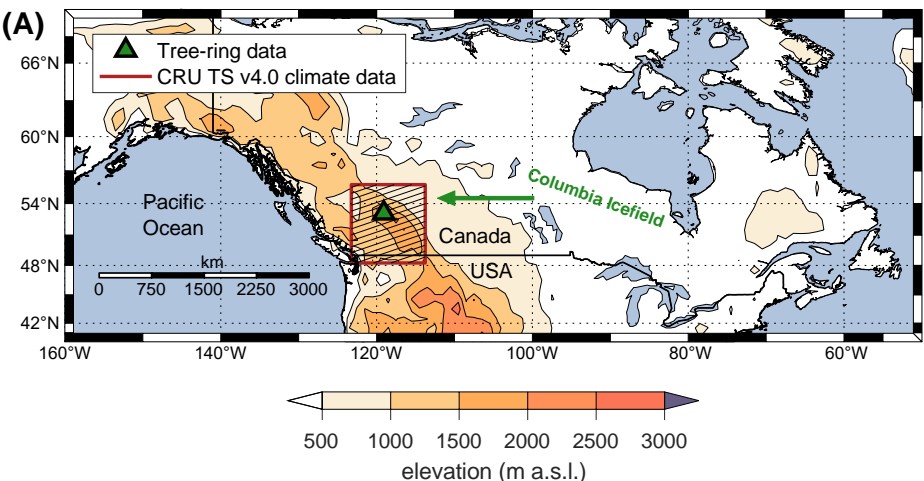




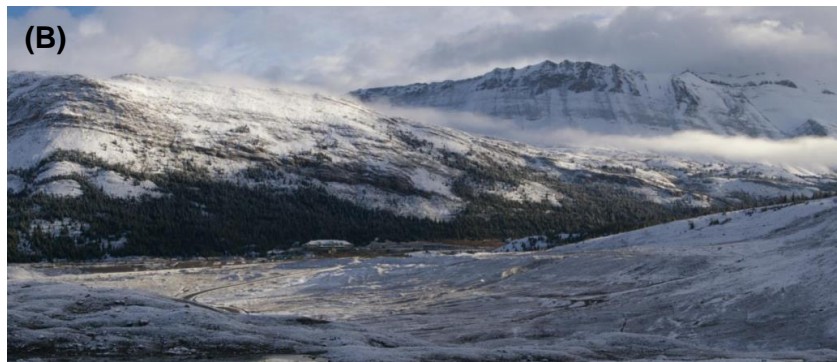


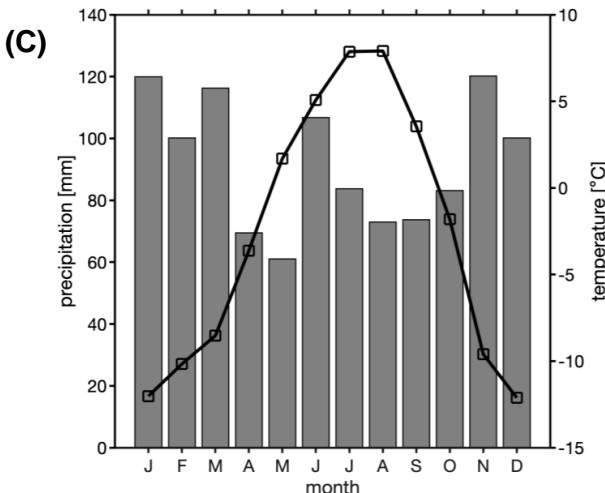


*Figure 1: A) Location of the Athabasca Glaciers at the Columbia Icefield, Canadian Rockies, where the wood cores for dendroanatomical measurements were collected in 2015. B) The Columbia Icefields site viewed from the Athabasca Glacier forefield, September 2018. The 2015 samples were obtained from sites east and west of the Icefields Centre (building located in the middle of the image). The Athabasca Glacier extended to the foot of the slope left in the photo in the 1840s. C) Monthly mean temperature and total precipitation (1970-2018 averages) for the CRU TS v4.03 grid point (52.25° N, 117.25° W) covering the Columbia Icefield area.*

Wood cores were washed in alcohol for 24 hours using a Soxhlet apparatus to remove resin and other soluble substances, and subsequently embedded in paraffin using a Tissue Processor TP1020 and Histocore Arcadia Embedding Center (Leica, Germany). A rotary microtome RM2245 (Leica Biosystems, Germany), equipped with N35 disposable microtome blades (Feather, Japan), were used to cut 12 μm thick transverse sections from the wood



cores. The thin-sections were stained with a 1:1 safranin-astrablue solution and mounted on
slides with Euparal (Carl Roth, Germany), following standard procedures (von Arx et al. 2016).
Digital images from each section were taken with a Zeiss Axio Scan Z1 (Carl Zeiss, Germany)
at a resolution of 2.3 pixels $\mu m^{-1}$. Tree-ring borders and individual tracheid cells were then
semi-automatically identified, and ring width as well as the position and anatomical dimension
of each tracheid cell were measured in the digital images using the image analysis software
ROXAS (v3.1) (von Arx and Carrer 2014). The anatomical parameters included cell lumen
area and cell wall thickness, where the latter was measured in four directions to obtain the
average cell wall thickness (CWT), i.e. two radial and two tangential cell walls per tracheid cell
(Prendin et al. 2017). Each tree ring was divided into 20 $\mu m$ wide bands parallel to the ring
border. In order to minimize the influence of outliers, the values corresponding to the 75$^{th}$
percentile within each 20 $\mu m$ wide band were computed. The anatomical density was derived
as the ratio of wall area to overall cell area (that is, including both wall and lumen area) in each
20 $\mu m$ wide band. Mork's index was used to separate the earlywood and the latewood portions
of the ring (Denne 1989). For further details regarding the dendroanatomical measurements,
see (Björklund et al. 2020).

*2.2 Chronology development*
From the potentially large number of possible dendroanatomical parameters, we narrowed
down subsequent analyses to seven parameters of anatomical dimensions, and three wood
density parameters based on anatomical dimensions, which are directly comparable to X-ray
and blue intensity-based microdensitometric parameters. The parameters are listed in table 1.
For comparative purposes, we also retained X-ray derived measurements of MXD (Luckman
and Wilson 2005), and the previously unpublished latewood blue intensity (BI) counterpart
(hereafter referred to as MXBI) measured on *P. engelmannii* from the Columbia Icefield area.
The X-ray MXD was produced using radiodensitometric techniques (Schweingruber et al.
1978) from 1.2-mm-thick laths, cut using a twin-blade saw along the tree cores but
perpendicular to the fiber direction (see Luckman and Wilson 2005 for details). For the
production of MXBI, the methodology outlined in (Rydval et al. 2014) was adopted. The MXBI
measurements were conducted using the CooRecorder software
(http://www.cybis.se/forfun/dendro/index.htm). Corresponding time series of ring-width were
also obtained and hereafter referred to as "original ring-width", as opposed to "ROXAS ring-
width", which were measured in program ROXAS on the fifteen cores used for the
dendroanatomical measurements. The X-ray MXD and MXBI datasets were originally
developed from living trees and snag material, however, to ensure consistency for the
parameter comparison, we used X-ray MXD, MXBI and original ring-width measurements from



living trees only (X-ray MXD: N = 78 series, MXBI: N = 182, and original ring width: N = 182,
see table 1). The dendroanatomical analysis was performed on tree cores for which original
ring-width and MXBI measurements were available. Thus, an additional subset based on the
fifteen trees was retained for the latter two parameters to ensure also a direct comparison with
the dendroanatomical chronologies. For the full MXBI dataset (N = 182), we additionally
derived eight partly overlapping percentile chronologies based on *absolute* ring-width, to
assess whether a similar ring-width dependence as previously reported by Björklund et al.
(2019) from Northern Fennoscandia could also be detected in the Icefields dataset, i.e. a ring-
width related differences of MXBI measurements taken in narrow versus wide rings. The
following ring-width percentile intervals were used: $0 - 30^{th}$, $10^{th} - 40^{th}$, $20^{th} - 50^{th}$, $30^{th} - 60^{th}$,
$40^{th} - 70^{th}$, $50^{th} - 80^{th}$, $60^{th} - 90^{th}$, and $70^{th} - 100^{th}$ to derive the sub-sampled MXBI
chronologies. Thus, for example, the $70^{th} - 100^{th}$ percentile chronology is computed from
MXBI-values measured in the 30% *widest* rings, while the $0 - 30^{th}$ percentile chronology
corresponds to MXBI-values from the 30% of the *narrowest* rings. Unfortunately, a similar
comparative analysis was not possible to conduct for the X-ray based MXD, as the
corresponding ring-width measurements originally developed, were unavailable to us in the
current study.
Since the analysis was performed on data derived from a cohort of living trees, capturing low-
frequency variability (i.e. decadal and longer) with RCS-type methods is a challenge (e.g.,
Briffa et al. 1992). Thus, we primarily focused here on the year-to-year (high-frequency)
signals in the tree-ring anatomical parameters, but still secondarily made tentative
observations of the lower frequency structures. To emphasize the interannual variations, the
individual dendroanatomical series were detrended in the program MATLAB (version
R2021a), by 1) fitting a cubic smoothing spline function with 50% frequency response cutoff
at 35 years to the raw tree-ring series (Cook and Peters 1981), 2) subtracting the fitted values
from the observed values to obtain detrended series (division was used to standardize the
ring-width measurements), and finally 3) averaging the detrended series by simple arithmetic
mean to produce the final parameter-specific chronologies (hereafter referred to as high-pass
filtered data). The same detrending procedure was performed on the MXBI, X-ray derived
MXD and original ring-width series, in order to obtain data that are comparable with the
dendroanatomical datasets. In addition, we also produced a set of non-detrended
chronologies by computing the arithmetic mean of raw time-series (hereafter referred to as
non-detrended data). All chronologies were truncated to the 1700-1994 period in the
subsequent analyses, to ensure a consistent overlap between datasets as well as a sufficient
sample depth ($N_{minimum}$ for the wood anatomical dataset = 9, $N_{maximum}$ = 15 cores for the 1700-
1994 period).


*2.3 Statistical methods*

To evaluate the strength of the between-series common signal and establish the replication needed to obtain mean chronologies meeting the commonly accepted standard, we used the RBAR (defined as the mean Pearson's correlation coefficient between all possible pairs of individual tree-ring series) (Wigley et al. 1984) and Expressed Population Signal (EPS) (Briffa et al. 1992) statistics. To assess the degree to which the various parameters co-vary, principal component analysis (PCA) and cross-correlations were computed over the 1700-1994 period.

Standardized tree-ring parameter chronologies were assessed for their relationship to regional monthly mean (Tmean) and maximum (Tmax) temperatures, by correlation against the monthly 0.5° x 0.5° gridded CRU TS v4.03 dataset (Harris et al. 2020) for the grid point average bounded by the latitude/longitude coordinates 48.25-55.75° N/113.75-123.25° W (Fig. 1). Tmax was included in the analysis because previous work has demonstrated slightly stronger calibration statistics than for Tmean when using MXD and ring-width chronologies for climate reconstruction in this region (e.g., Heeter et al. 2021; Wilson et al. 2019; Wilson et al. 2014; Wilson and Luckman 2003). The associations with monthly precipitation totals and minimum temperatures were also tested, but not included here due to weak significant empirical relationships. The lack of precipitation sensitivity of *P. engelmanni* in the Icefield area was already noted in George and Luckman (2001) which is not surprising as the trees are growing in temperature limited upper tree-line environments. To make the climate sensitivity analysis comparable to previous studies from the Columbia Icefield area, we also included the homogenized (1895 – present) 50 x 50 km gridded temperature data originally developed by the Meteorological Service of Canada and previously used in Luckman and Wilson (2005) to reconstruct last-millennium summer temperatures for the Canadian Rockies. Similar to Luckman and Wilson (2005), we used the mean of four grids closest to the Columbia Icefield area. Calibration trials with these data are provided in the supplement (fig. S1 and S2).

Further, the dynamic nature of the temperature signal (i.e. optimal target season and its temporal stability) was evaluated through moving window correlation analysis between tree-ring chronologies and daily temperature data (grid 52.5° N, 118.5° W) from the Berkeley Earth dataset (http://berkeleyearth.org/data/) (Rohde and Hausfather 2020) covering the 1880 – recent period. Pearson's correlations were computed for 30-year sliding windows with a 1-year offset. For each 30-year block, temperatures were averaged in 30-day long windows which were shifted at daily time steps throughout the year (sensu Jevsenak and Levanic 2018). To ensure the analysis was not affected by long-term trends, the temperature data were high-pass filtered prior to analysis using the same 35-year filter as was used to detrend the tree-ring parameters.



## 3. Results and discussion

*3.1 Picea engelmannii dendroanatomy characteristics*

Besides the conventional width parameters (i.e., ring width, earlywood- and latewood width, referred to as "ROXAS" in table 1), seven anatomical parameters and three anatomically-based density parameters, measured from 15 cores and covering the period 1585 – 2014 CE, were retained for analysis (see table 1). Basic chronology assessment (table 1) shows, in line with previous studies on temperature-sensitive conifers (Björklund et al. 2020), that maximum radial cell wall thickness (Max. radial CWT) and anatomical MXD (aMXD) are the two anatomical parameters with the highest mean inter-series correlation (RBAR = 0.47 and 0.48, respectively). For both parameters, EPS reaches the 0.85 threshold (Wigley et al. 1984) with 6 series (table 1). Notably, these values are of comparable strength to the RBAR and EPS of X-ray based MXD (RBAR = 0.49, 6 trees required for EPS = 0.85). By comparison, the RBAR for MXBI is surprisingly low at 0.19 and the replication needed to attain the EPS of 0.85 is 24 series. These MXBI chronology statistics are lower than for ring width (RBAR = 0.22 and 0.28 for original and ROXAS ring width, respectively) – an observation noted previously by (Rydval et al. 2014; Wilson et al. 2019). The RBAR and EPS values for MXBI slightly decrease if computed only on the 15 trees that have been pre-selected for the dendroanatomical analysis. This is surprising given that the selection of the cores for dendroanatomy was partly based on its ring-width signal strength (see sect. 2.1), and that the RBAR and EPS statistics for ring width actually improve when narrowing the analyses down to these 15 trees (see table 1). Although the BI-based density parameters typically require a larger sample size than ring width (e.g., Blake et al. 2020; Wilson et al. 2021) for a robust chronology, the MXBI chronology statistics obtained for *P. engelmannii* from our site are still lower than the previously reported MXBI findings for the same species across British Columbia, Canada (Wilson et al. 2014).

Notably, several anatomical and density parameters are found to exhibit a rather low common signal, yet a reasonably strong temperature sensitivity (see sect. 3.2). These include, in decreasing order of signal strength: earlywood (EW) cell wall area (RBAR = 0.13), EW lumen area (RBAR = 0.12), EW density (RBAR = 0.10), EW cell area (RBAR = 0.09) and latewood (LW) cell area (RBAR = 0.09). The replication required to attain a robust EPS ranges between 38 (EW cell wall area) to 57 trees (EW cell area and LW cell area).

**Table I:** *Basic summary statistics for each high-pass filtered parameter chronology. Abbreviations used in the table are EW (earlywood), LW (latewood), CWT (cell wall thickness), aLWD (anatomical latewood density) and aMXD (anatomical maximum latewood density). Parameters highlighted in grey are those requiring the lowest sample replication to reach an EPS above the arbitrary threshold level of 0.85.*


| | # samples | RBAR | *n* for EPS (0.85) |
|---|---|---|---|
| **Width parameters** | | | |
| Original ring-width (from Luckman 1997; Luckman and Wilson 2005, and later unpublished updates) | 182 | 0.22 (0.27 for N = 15)* | 20 (15 for N = 15)* |
| ROXAS ring-width | 15 | 0.28 | 15 |
| ROXAS EW width | 15 | 0.26 | 16 |
| ROXAS LW width | 15 | 0.19 | 24 |
| **Earlywood anatomy** | | | |
| EW cell area | 15 | 0.09 | 57 |
| EW Lumen area | 15 | 0.12 | 42 |
| EW cell wall area | 15 | 0.13 | 38 |
| **Latewood anatomy** | | | |
| LW cell area | 15 | 0.09 | 57 |
| LW Lumen area | 15 | 0.31 | 13 |
| Max. radial CWT | 15 | 0.47 | 6 |
| Max. tangential CWT | 15 | 0.34 | 11 |
| **Density parameters** | | | |
| EW density | 15 | 0.10 | 51 |
| aLWD | 15 | 0.28 | 15 |
| aMXD | 15 | 0.48 | 6 |
| MXBI (unpublished) | 182 | 0.19 (0.16 for N = 15)* | 24 (30 for N = 15)* |
| X-ray MXD (from Luckman and Wilson (2005)) | 78 | 0.49 | 6 |

*the RBAR and EPS values in parentheses are for the original ring-width and MXBI time-series
computed for exactly the same 15 trees that have been used to produce the wood anatomy datasets.

The co-variability between the various parameters over their common 1700-1994 period was
assessed through principal component analysis and cross-correlations (fig. 2). The first two
components (PC1 and PC2) express a cumulative 68.1% of overall variance amongst the
datasets. The PC1 alone explains 43.8% of variance, and is dominated by latewood-related
parameters, including both anatomy and density parameters. We found that aMXD, Max.
radial CWT and X-ray MXD cluster together in the bivariate plot, showing that all three
parameters express comparable signals (also corroborated by the correlation matrix in fig.
2b). The MXBI also loads strongly positively on PC1, but slightly separates from this cluster
by being positively correlated to PC2. Among the LW density-related components, MXBI is
the parameter best correlated with ring-width and latewood-width chronologies (fig. 2b),
although these correlations are only moderate (r $_{MXBI\ vs.\ original\ ring\ width}$ = 0.43, r $_{MXBI\ vs.\ latewood\ width}$
= 0.66). The principal component analysis including the subsampled MXBI percentile
chronologies based on the *absolute* corresponding ring widths reveal that the correlation
coefficients against the latewood width, and to some degree also ring width, successively

 

increase for the "narrow-ring MXBI chronologies" (fig. S4). The "wide-ring MXBI chronologies"
(i.e., ~50th-100th percentiles) are, on the other hand, more similar to the aLWD, Max. radial
CWT, aMXD and X-ray MXD chronologies. This observed ring-width inclination of MXBI
suggest that the dataset might be subject to a resolution bias (Björklund et al. 2019). More to
this potential issue in sect. 3.3.

The variance of PC2 (24.3 % of total variability) is dominated by ring width and earlywood-
related density and anatomy parameters. Amongst these, EW density stands out by loading
strongly negatively on the PC2 axis (reflecting its negative association with early-summer
temperatures, see sect. 3.2). Moreover, the EW cell wall area stands out by loading more
strongly on the PC1 axis than on the PC2 axis, and by clustering more closely with the
latewood than with the earlywood components (reflecting its late-summer temperature
sensitivity, see sect. 3.2).

In summary, the PCA results suggest a high degree of shared signal amongst the datasets.
As we detail further in the next section, PC1 is dominated by variables showing a pronounced
late-summer (July-August) temperature sensitivity, while variables loading on PC2 are those
that most strongly correlate with mid-summer (June-July) temperatures.

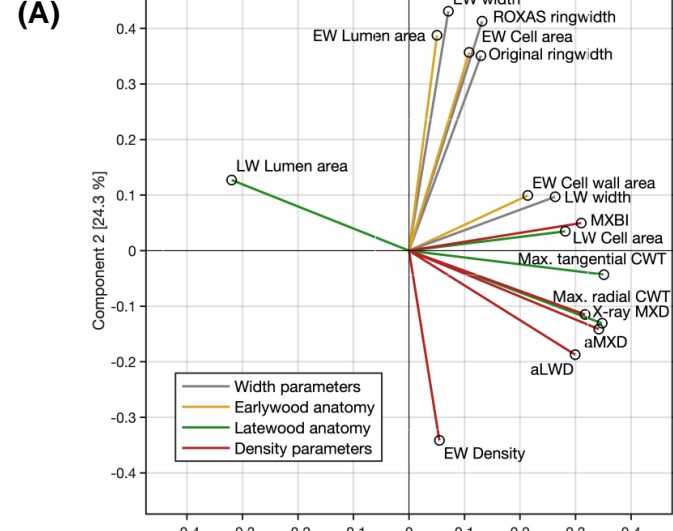




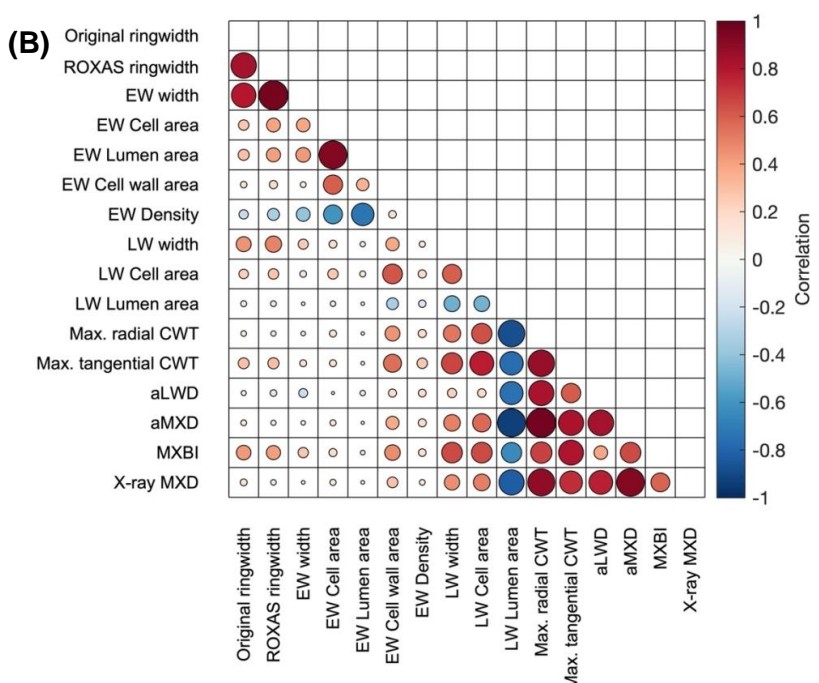


**Figure 2:** *A) biplot of the first two principal components of the PCA performed 1700-1994 CE period on the width, anatomy and density parameters. The colors of the vectors correspond to the parameter grouping used in table 1. The first two components together represent nearly 68% of the total variation. B) correlation between various anatomical and width parameters. X-ray MXD and MXBI are included for comparison. Correlations are computed over the common 1700-1994 period using high-pass filtered chronologies. The color and size of the markers denote the direction and strength of the relationships.*

*3.2 Climate response*

Simple linear correlations between selected parameters and monthly CRU TS mean (Tmean) and maximum (Tmax) high-pass filtered temperatures are shown in fig. 3. In line with previous work from North America (Harley et al. 2021; Heeter et al. 2021; Luckman and Wilson 2005; Wilson et al. 2014; Wilson and Luckman 2003), our results reinforce the importance of Tmax temperatures for wood formation and growth of *P. engelmannii* in the region by providing, in general, slightly higher correlation values for Tmax than for Tmean. Interestingly, the pattern observed in North America contrasts to many other temperature-limited regions of the Northern Hemisphere, where conifers have generally been noted to correlate stronger to Tmean than to Tmax (observation made by the author team, results not published). Whether



this is actually grounded in a tree physiological mechanism is still an open question. Furthermore, the general pattern revealed by the climate response analysis shows that the various dendroanatomical traits respond to consecutive temporal windows within a short seasonal window extending from June to August, in line with our understanding of the successive physiological processes (i.e., cell expansion and cell wall thickening) behind wood formation and growth (e.g., Fonti et al. 2013). These results support the climate-response pattern that has generally been observed for conifers across the Canadian Rockies (Luckman and Wilson 2005) and the adjacent Interior British Columbia (Wilson et al. 2014; Wilson and Luckman 2003). Even though the parameters describe two temporally distinct temperature signals, both are encapsulated within the short June-July-August period. The narrow window of response patterns is most likely constrained by the distinct and short warm season characterizing the climatology of the study site, where average monthly temperatures rise above 0 °C only in four months of the year (fig. 1c). This window is substantially shorter than the single but wide target season observed in the latewood anatomical traits of *P. sylvestris* growing in temperature-limited environments in northern Scandinavia (Björklund et al. 2020).

The anatomical properties of earlywood, as well as ring width, in general respond to peak-summer temperatures (June and in some cases also July). Earlywood (EW) density displays significant ($p < 0.01$) albeit weak sensitivity, expressed through a negative correlation with June temperatures and a positive correlation with July temperatures. These results broadly agree with the large-scale tendency observed previously across the Northern Hemisphere temperature sensitive conifer density network (Björklund et al. 2017). Lumen area displays a similar yet opposite pattern, i.e. a positive (negative) correlation with June (July) temperatures. Although the opposite patterns in EW density and EW lumen area are intuitive since low earlywood density is mechanistically connected to a large lumen area, the switch in sign of the signal within each parameter is more difficult to interpret. In this context, it is noteworthy that the target season for earlywood cell wall area differs from the general pattern of the earlywood in that the strongest, albeit insignificant correlations, are shifted towards the July-August season. This is also evident from the PCA biplot in fig. 2. However, the monthly correlation pattern of this parameter is actually broadly inverse to that of EW lumen area, supporting the notion that lumen area and cell wall area are just two sides of the same coin. When lumen area is larger, the cell wall area is conversely relatively reduced, but how temperature drives the intricate intra-annual development remains unknown. This, in turn, complicates their potential use as climate proxies.



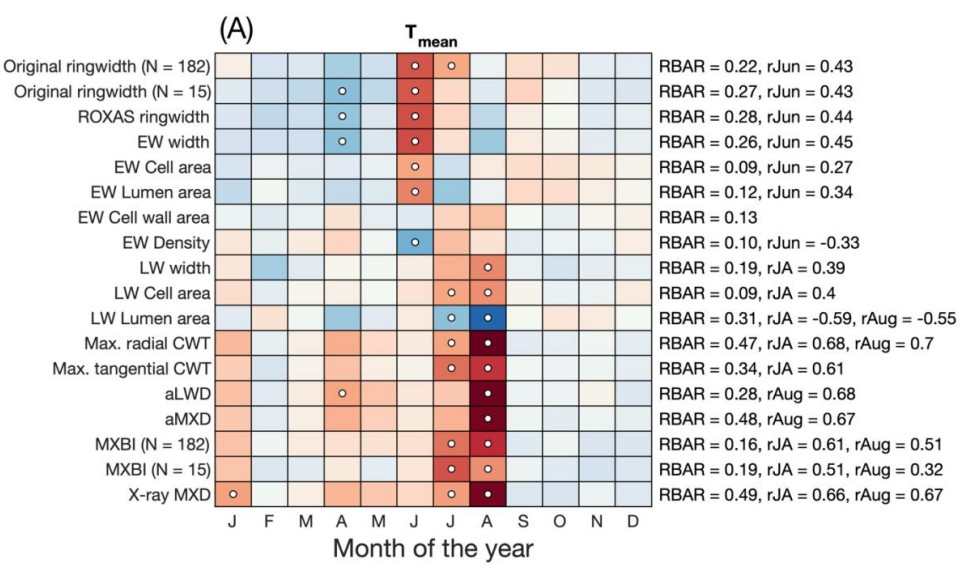

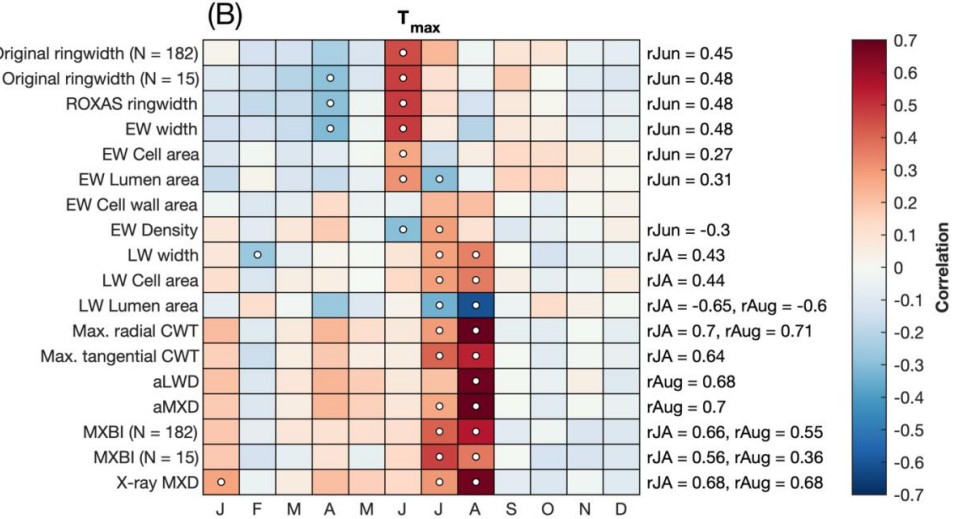


***Figure 3:*** *Correlations between tree-ring parameters and monthly (A) average ($T_{mean}$) and (B)*

*maximum ($T_{max}$) temperatures from the CRU TS v4.03 product (48.25-55.75° N/113.75-*
*123.25° W subset average). Correlation coefficients are computed over the 1901-1994 period*
*using high-pass filtered data. The RBAR statistics for each parameter chronology, and*
*correlation coefficients with seasonally averaged temperature are provided on the right side*
*of the plots. For original ring width and MXBI, results are also provided for chronologies*
*(denoted as N = 15) built from the same 15 trees that are used to produce the dendroanatomy*
*data. Significant correlations (p<0.01) are marked with white circles. Correlations with*



*temperature data produced by the Meteorological Service of Canada are provided in the*
*supplement (fig. S1).*

Focusing on the anatomical components of the latewood, the dominant temperature signal shifts to late-summer, predominantly August. Parameters displaying the strongest temperature sensitivity are also those showing the highest RBAR statistics (table 1) – that is, aMXD and Max. radial CWT. The imprints of high-frequency temperature variability within these two parameters are, however, very similar, if not identical, to that of the MXD derived from the X-ray technique. By comparison, the exceptionally weak inter-series signal strength of the MXBI parameter (table 1) is compensated by high replication (N = 182), and thus MXBI is also rather similar to aMXD, Max. radial CWT and X-ray MXD. However, the temperature signal of MXBI is shifted earlier by expressing stronger correlation with July temperatures but weaker with August compared to aMXD, Max. radial CWT and X-ray MXD. The aggregated July-August temperature response of MXBI is thus in fact only marginally weaker than that of X-ray MXD, aMXD and Max. radial CWT.

The reason why the monthly correlations of the full MXBI dataset (N = 182) differ slightly from the more physically direct density and anatomy parameters could be related to the lower measurement resolution that artificially makes it more similar to ring width and latewood width (Björklund et al., 2019). Recall that the cross-correlation (fig. 2) and the PCA biplot based on the percentile MXD chronologies (fig. S4) confirmed this enhanced relationship with ring width/latewood width. To test this theory further, we have in figure 4 correlated the percentile MXBI chronologies against the target July-August Tmax (fig. 4a) and against the full (N = 182) high-pass filtered original ring-width chronology (fig. 4b), using resampling of data. Unfortunately, corresponding latewood width measurements are not available for MXBI, so this comparative analysis is restricted to ring width. Nevertheless, we find that when using the full July-August season the poorest temperature imprint is found in the MXBI values of the narrowest (~40%), *and* the widest (~40%) of the rings, while the strongest July-August signal can be recovered from the MXBI-values in rings that are close to average in width ($40^{th} - 70^{th}$ percentile). Expanding the climate correlation analysis to monthly Tmax data (fig. 4c) reveals, however, a gradual transition from predominantly an August temperature signal in the wide ring MXBI chronologies towards being more dominated by a July signal in the narrow ring MXBI chronologies. MXBI-values in rings that are close to average in width correlate equally strong to both July and August, which explains the overall better performance of these data when comparing to the July-August target (fig. 4c). Importantly, we find no correlation between the MXBI and ring-width in the widest rings. However, as we move towards narrower rings, the MXBI-values becomes successively more alike the ring width/latewood width (fig. 4b and





fig. S4). All in all, these results suggest that an effect of low measurement resolution may be
present for narrower ring widths/latewood widths. If so, this means that the MXBI parameter
may become subject to greater target seasonal uncertainty, which may fluctuate between July
and August signals through time, largely depending on the absolute ring width/latewood width
of the analyzed tree-ring sample collection.

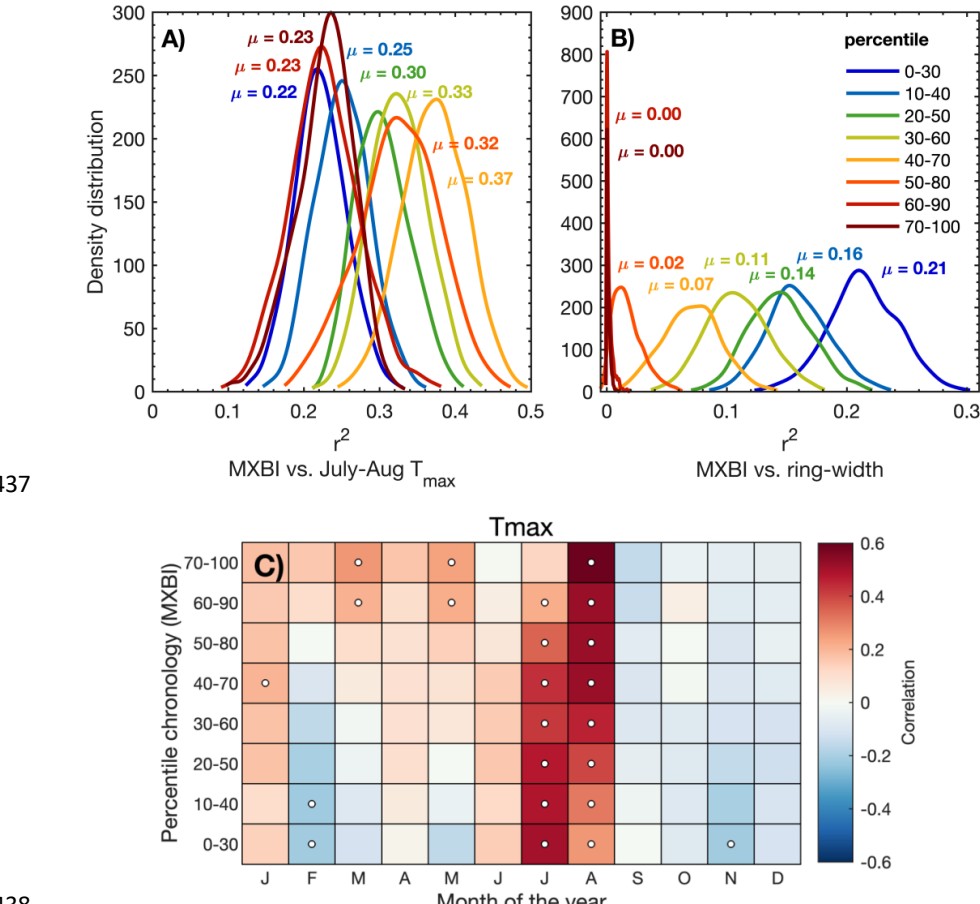


***Figure 4:*** *A)-B): The density distribution of r2-values obtained from 1000 calibration trials*
*(1901-1994 period) where MXBI chronologies are built from 100 series randomly drawn from*
*the total of 182 series without replacement. The high-pass filtered MXBI values are sorted into*
*percentiles based on the absolute ring-width (e.g., the 0-30 percentile are the corresponding*
*MXBI-values for the narrowest 30% of the rings), and then averaged into percentile*
*chronologies. A) the calibration r2-values between these chronologies and high-pass filtered*
*July-August CRU TS Tmax, B) same as A) but calibrated against the full (N = 182) high-pass*
*filtered ring-width chronology. C) Correlation between the MXBI percentile chronologies and*
*monthly maximum (Tmax) temperatures from the CRU TS v4.03 product (48.25-55.75°*





*N/113.75-123.25° W subset average). Correlation coefficients are computed over the 1901-*
*1994 period using high-pass filtered tree-ring and temperature data. Significant correlations*
*(p<0.01) are outlined with white circles.*

Focusing only on anatomical traits with the highest temperature sensitivity (aMXD and Max. radial CWT), comparison against daily temperatures (Fig. 5) confirms a significant and strong mid/late summer signal over the 1880-1994 period. Breaking down the climate response in daily increments reveals that the strongest signal (r > 0.5) occurs on average between day 192 and day 251 of the year (i.e. July 11[th] until September 8[th]-9[th], with a peak correlation of 0.73 and 0.74 for Max. radial CWT and aMXD, respectively, occurring between 21[st] of July-20[th] of Aug and 23[rd] of July-22[nd] of August). The temperature associations at the peripheral ends of the target season are, however, more elusive. We note, for example, that the September signal disappears around the first half of the 20[th] century for both anatomical parameters. However, the Berkeley Earth gridded daily temperature dataset used herein is at this stage considered experimental (see http://berkeleyearth.org/data/). Some of the correlation structure observed in figure 5 can thus be related to climate data quality rather than to the characteristics of the proxy datasets. Nevertheless, a similar correlation structure holds for X-ray derived MXD and to a lesser degree MXBI (N = 182), but the two parameters exhibit enhanced correlation coefficients in the second half of the 20th century compared to the early period (also corroborated by the split-period calibration in figure 6). Moreover, despite the high sample replication, MXBI shows slightly weaker correlations with daily data than the other density-related parameters, particularly in the early 1880-1930 period, when ring widths coincidentally are the narrowest in the record (see fig 7). For comparative purposes we also include anatomically derived ring width, which shows, on average, the strongest correlations (r = 0.3 to 0.5) with temperatures between day 146 and 206 of the year (i.e. May 26[th] to July 25[th]).

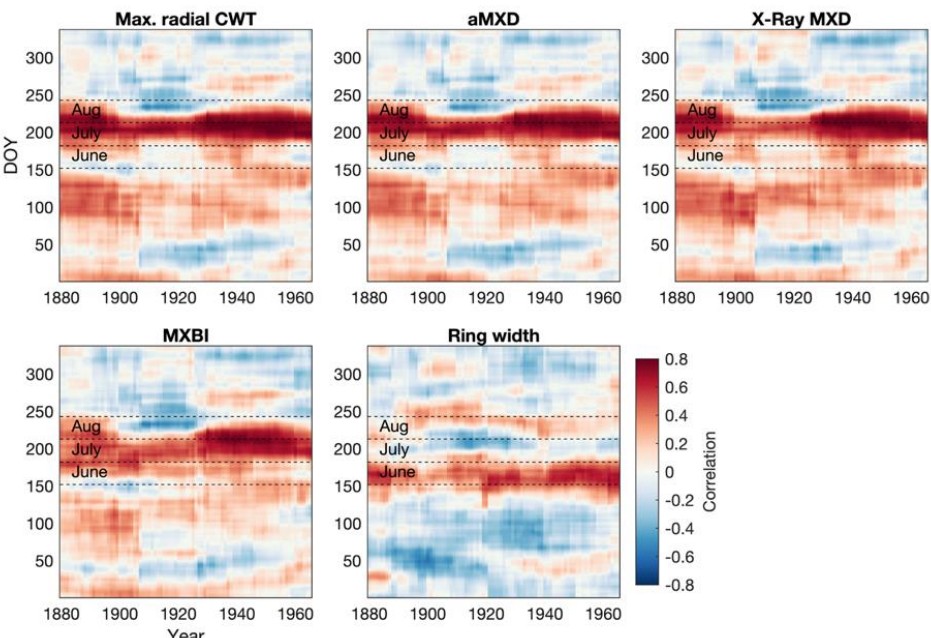

474

**Figure 5:** *Moving correlation between the full tree-ring parameter datasets and Berkeley Earth gridded daily temperatures (grid 52.5 °N. 118.5°W, 1880-1994 period). A 30-year moving window, shifted by one year, was used in the analysis. Temperatures were averaged over a 30-day window, and shifted throughout the year at daily steps. The days on the x- and y-axis thus show the first day of the 30-year and 30-day windows, respectively. E.g., day 152 on the y-axis represents the period from June 1 to June 30. Both tree-ring and temperature data have been high-pass filtered prior to analysis. The June-August season is highlighted to aid interpretation.*

483

The stability of the July-August temperature signals of aMXD and Max. radial CWT, along with X-ray MXD and MXBI, were further assessed by a split-period calibration procedure, where the full instrumental period 1901-1994 was split into two subperiods of equal length (1901-1948 and 1949-1994) (fig. 6). Calibration trials were performed on the high-pass filtered tree-ring and CRU TS temperature datasets, but also on non-detrended time-series to evaluate the influence of the long-term trends. The two wood anatomical parameters calibrate more strongly to the early period compared to the late, both when using Tmean and Tmax. However, especially for Max. radial CWT, the calibration differences in the two periods are slight ($R^2$ = 53% and 47% against Tmax for the 1901-1948 and 1949-1994 periods, respectively). By comparison, the X-ray MXD calibrate more strongly in the latter half of the instrumental period and show more pronounced temporal instabilities ($R^2$ = 34% and 55% against Tmax for the



1901-1948 and 1949-1994 periods, respectively). This contrasts to the prior finding (Luckman
and Wilson 2005), where no such instabilities in the early 20[th] century were detected. These
contrasting results are most likely not related to using different climate data products because
similar results (fig. S2) were obtained when using the Luckman and Wilson (2005) temperature
data, originally produced by the Meteorological Service of Canada. Instead we suspect that
the discrepancy can be attributed to either using a larger network of MXD data than used in
this study, or that Luckman and Wilson (2005) used multivariate regression models (including
ring width and lagged growth responses) to explain a wider target season than attempted here.

Calibration trials with high-pass filtered data over the full period 1901-1994 reveal that Max.
radial CWT performs overall best (Tmax $R^2$ = 49%), closely followed by aMXD ($R^2$ = 0.46%)
and X-ray MXD ($R^2$ = 0.46%). The temporal instability of X-ray MXD and by comparison the
robust and strong signals of the aMXD and especially the Max. radial CWT parameters are
further confirmed by the resampling calibration trials presented in fig. 6c, where 10 random
series are drawn from the sample cohorts 1000 times without replacement, and the resulting
parameter chronologies are subsequently correlated against July-August Tmax. The reason
for the X-ray MXD loss in signal is difficult to disentangle, but it is unlikely related to having
different samples for the X-ray and anatomical datasets because the resampling scheme
clearly show that the $r^2$-distributions are different (fig. 6c). We note however that the
correlations between the various latewood parameters against ring widths change from the
early to late 20[th] century periods, *and* that the correlations slightly differ in magnitude and sign
(fig. 7). We find that that MXBI is positively correlated width ring width, whereas the
correlations for X-ray MXD range between non-significant to weakly positive. The Max. radial
CWT, on the other hand, show a non-significant or weak negative correlation with ring width
during the 20[th] century. This gradual, and slightly larger shift in moving window correlation
against ring width during the early 20[th] century may thus be an indication that both MXBI and
X-ray MXD are challenged by comparatively low measurement resolution. This clearly needs
further scrutiny because it may be important for the interpretation of inferred climate signals
back in time, particularly because the ring-width correlation converges for the X-ray and
anatomy data but dramatically diverges for MXBI. The lower late-period signal of the
anatomical parameters compared to X-ray MXD requires a different explanation. According to
the distribution of the $r^2$-values in the resampling scheme of figure 6c, the late period Tmax
signals are not appreciably different, so perhaps this is simply by chance compounded with
having five times higher X-ray MXD replication.

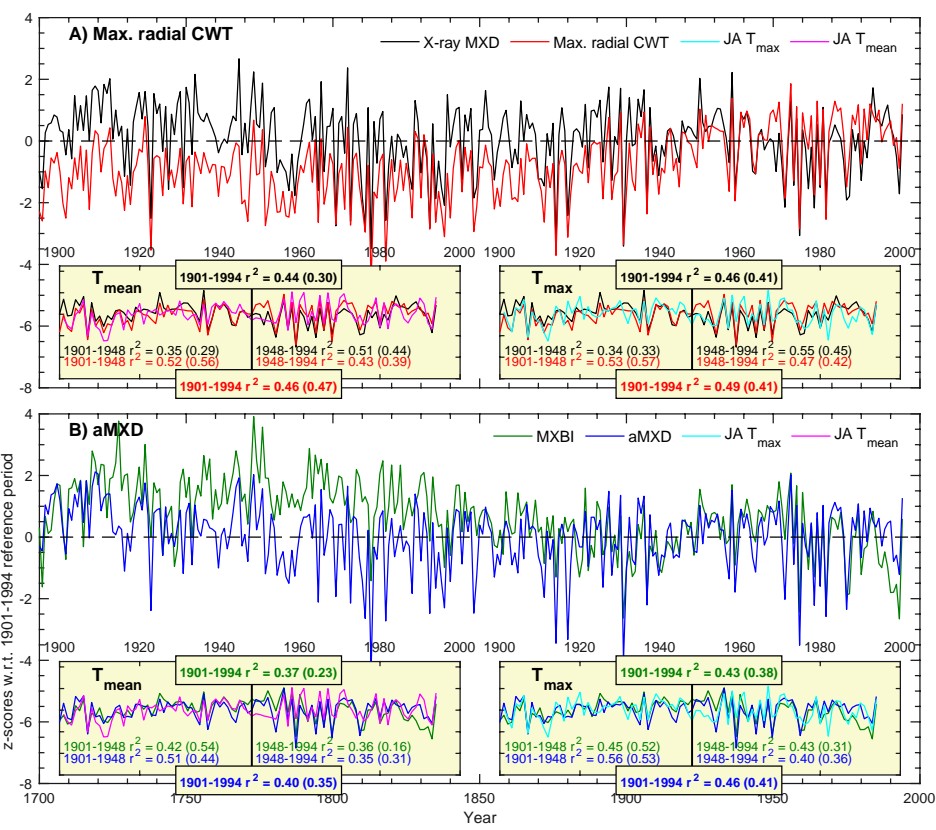


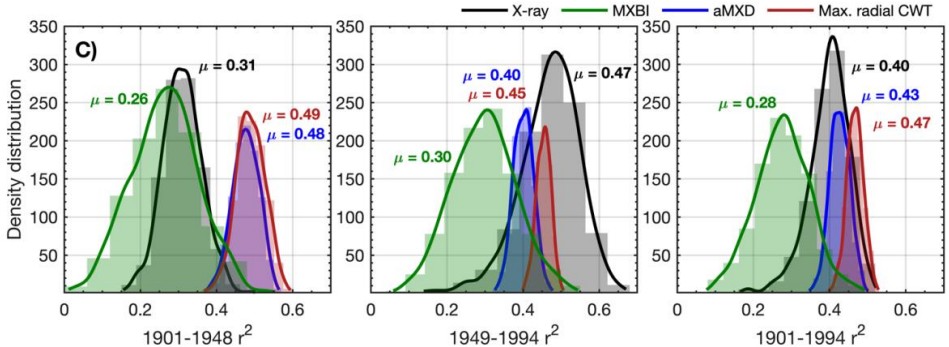


**Figure 6:** *A-B): Full (1901-1994) and split-period (1901-1948, 1949-1994) calibration statistics for the Max. radial CWT (red line), aMXD (blue line), X-ray MXD (black line) and MXBI (green line) chronologies against July-August mean and maximum CRU TS temperature. Coefficients of determination ($r^2$) are provided both for high-pass filtered and non-detrended mean data, where the latter are shown in parentheses. Time-series in the figures show non-detrended*



*mean chronologies, z-scored over the instrumental 1901-1994 period. C): The density*
*distribution of r2-values obtained from 1000 calibration trials where parameter chronologies*
*are built from 10 series randomly drawn without replacement from the sample cohort. The*
*resampling trials are based on high-pass filtered climate and tree-ring data. Calibrations are*
*performed against July-August maximum temperatures.*

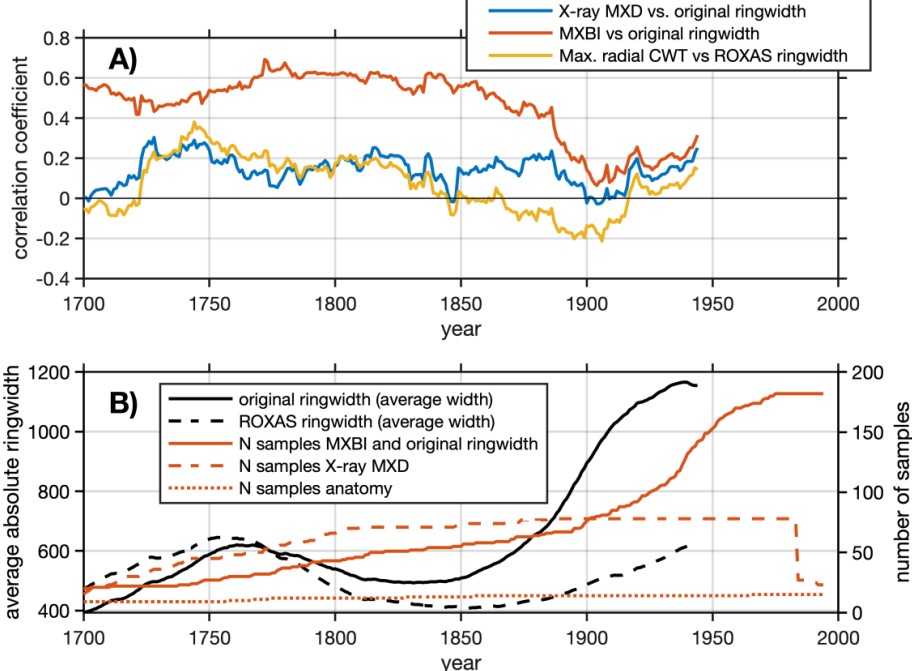


**Figure 7:** *A) running correlation (a 50-year window shifted by one year) between selected*
*density parameters and ring width. The years on the x-axis show the first year of the 50-year*
*correlation windows. Note that for X-ray MXD, the ring-width data are not obtained from the*
*same tree cores as have been used for the density measurements, which is otherwise the*
*case for both MXBI and anatomy. B) running average of absolute ring widths (original and*
*ROXAS datasets) computed using a 50-year window shifted by one year, together with the*
*chronology sample depths of the X-ray MXD, MXBI and dendroanatomical datasets.*

*3.3 Long-term trends*
Justification of the cost and time constraints currently associated to the production of long
dendroanatomical datasets requires that there must be an information gain not obtainable
from conventional techniques. In fact, high-resolution, cell-based, measurements already offer
an advantage when it comes to the understanding of the structure – function relationships
(e.g., Bouche et al. 2014; Pittermann et al. 2011; Wilkinson et al. 2015), the complex



mechanisms behind tree-ring formation (Rathgeber et al. 2016), with relative timestamps
(Ziaco 2020) of brief intra-seasonal climate extremes, such as late growing season cold spells
or initiation of volcanic cooling episodes (Edwards et al. 2021; Piermattei et al. 2020). The
question remains, however, whether dendroanatomy can also provide additional paleoclimate
information, in particular across multi-decadal and longer frequencies. If MXBI, and, perhaps,
to a lesser degree X-ray MXD, are challenged by lower measurement resolution, muting the
inter-annual climate signal when ring (latewood) widths are narrow, this dependence could
affect the lower frequencies, and introduce an inflated multi-decadal variability (Esper et al.
2015). Moreover, the fidelity to the monthly temperature targets may exhibit instability when
rings (latewoods) are narrow, shifting back and forth between August or July dominated
signals (exemplified in fig. 4c). It is at the moment unclear how this phenomenon could affect
the lower frequencies of our chronologies. Moreover, periods with persistence in narrow ring
widths will force MXBI, and perhaps also X-ray MXD, to exhibit persistently low densitometric
values (Björklund et al. 2019). Exacerbating this issue is that persistently narrow ring
width/latewood width may not even be a product of the distinct and earlier temperature target
(June-July, fig. 3), but could also be related to stand dynamics/disturbances (Rydval et al.
2018), and thus pass down non-climatic distortions of decadal to centennial variations to X-
ray MXD and MXBI. The anatomical parameters may not be perfect, however, as part of a
multi-parameter approach they can serve to evaluate the potential risk of a resolution bias (in
X-ray MXD and MXBI) when implementing these parameters both on shorter and longer
timescales.
A robust picture of long-term trends in dendroanatomical parameters can only emerge from
analysis of millennial length, multi-generation, composite chronologies suitable for RCS-type
analysis (Briffa and Melvin 2008). However, by exploring corresponding parameters derived
using different techniques we can already make some tentative conclusions. Figure 8 shows
average non-detrended time series of selected tree-ring parameters, z-scored over the 1901-
1994 reference period. We find contrasting long-term trends in most of the selected tree-ring
parameters, as well as a varying prevalence of extremely high or low single-year values (see
also the probability density functions in fig. 9). In fact, the only two parameters with somewhat
comparable secular trends – the X-ray MXD and the aMXD, display minimal (non-significant)
long-term linear trends over the 1700-present period. As for the modern period, neither of the
two parameters show any significant linear trend. While it might be tempting to draw parallels
to the regional warm-season (July-August) Tmax record, which lacks any significant linear
trend (lower panel fig. 8), we do not yet have sufficient evidence to determine if this is indeed
a signal-related proxy feature or an artifact caused by tree age. The trees used to produce the
wood anatomical datasets come from more or less the same age class. Thus, even if the tree





rings are cambial-age aligned (such as in fig. 9), it is problematic at this point to determine if
this is an age-trend. The reason for the trend mismatch between aMXD and Max. radial CWT
is difficult to assess, but we can reject that it is a product of a resolution bias. If it simply is
related to differences in age-related trends, we can only resolve this issue when more samples
with a broader temporal distribution are available for analysis. Nevertheless, at this early
stage, it is worth mentioning that Max. radial CWT performs better than any other tested
parameter in the full calibration procedure using unfiltered data in fig. 6, particularly for *mean*
July-August temperatures.

The mean chronology of MXBI shows a steady significant decrease in values through time.
Although *P. engelmannii* is, in general, characterized by light-colored wood that has few
discoloration artifacts (Heeter et al. 2020; Wilson et al. 2014), the negative trend seen here is
likely, at least partly caused by the transition in color at the heartwood – sapwood boundary,
previously shown to bias the BI-measurements of various species (Björklund et al. 2014;
Rydval et al. 2014). However, it may again also be partly related to the lower measurement
resolution (see fig. 4). We particularly draw attention to the recovery in MXBI values around
the 1900s coinciding with the time when ring width values also starts to increase from an
extended decline. The break in the positive trend of MXBI, covering 1900-1960 CE, again
coincides with a slight dip in the ring width (cf. dotted line in the ring width panel of fig. 8).



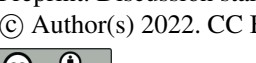

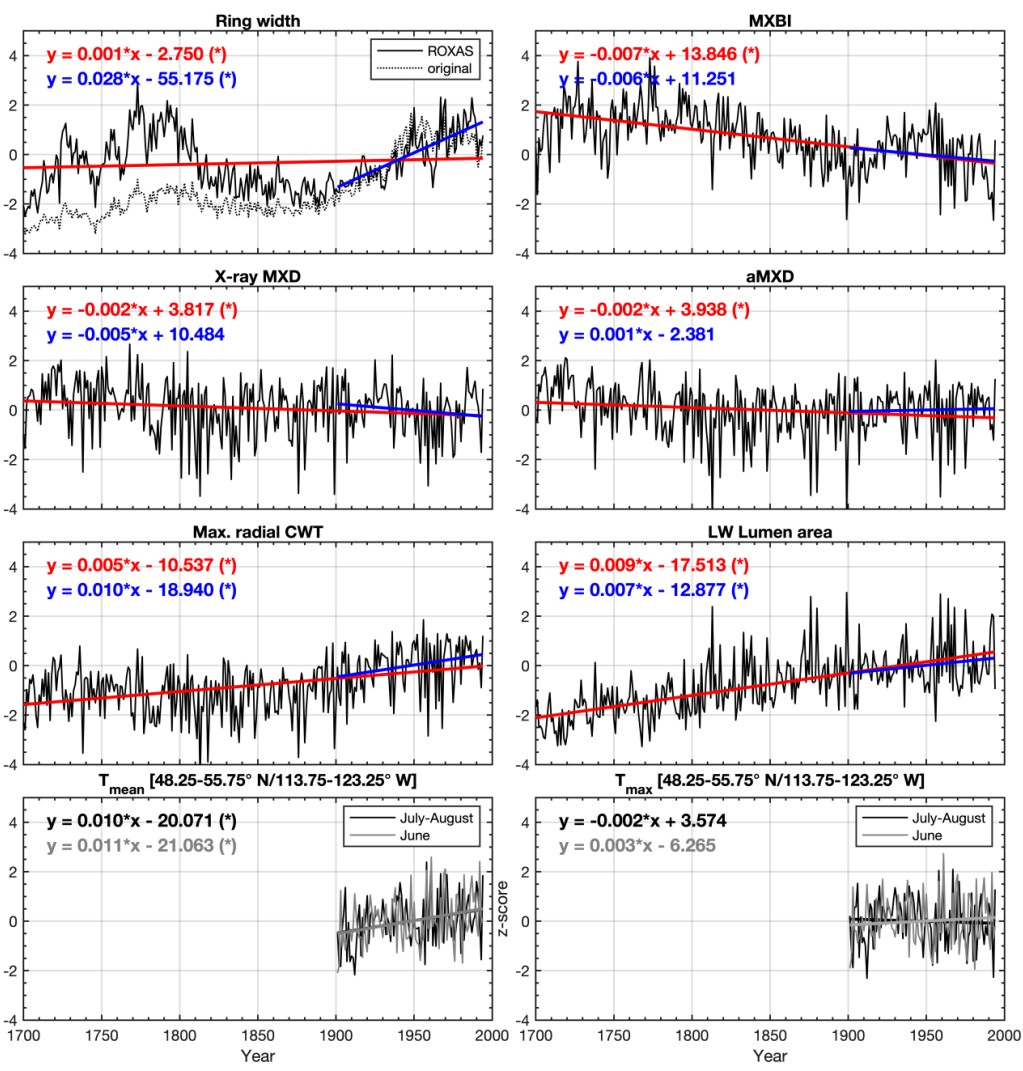

**Figure 8:** *Average non-detrended time series of selected tree-ring parameters, z-scored over the 1901-1994 reference period. The blue and red lines show the linear trends over the 1901-1994 and 1700-1994 periods, respectively. For ring width, the trends are computed only for the series used for the anatomical analyses. Seasonally averaged June-August (48.25-55.75° N/113.75-123.25° W CRU TS v4.03 subset average) mean ($T_{mean}$) and maximum ($T_{max}$) temperatures are provided for comparison. (\*) indicates a significant trend ($\alpha = 0.05$) estimated by the Mann-Kendell trend detection test.*



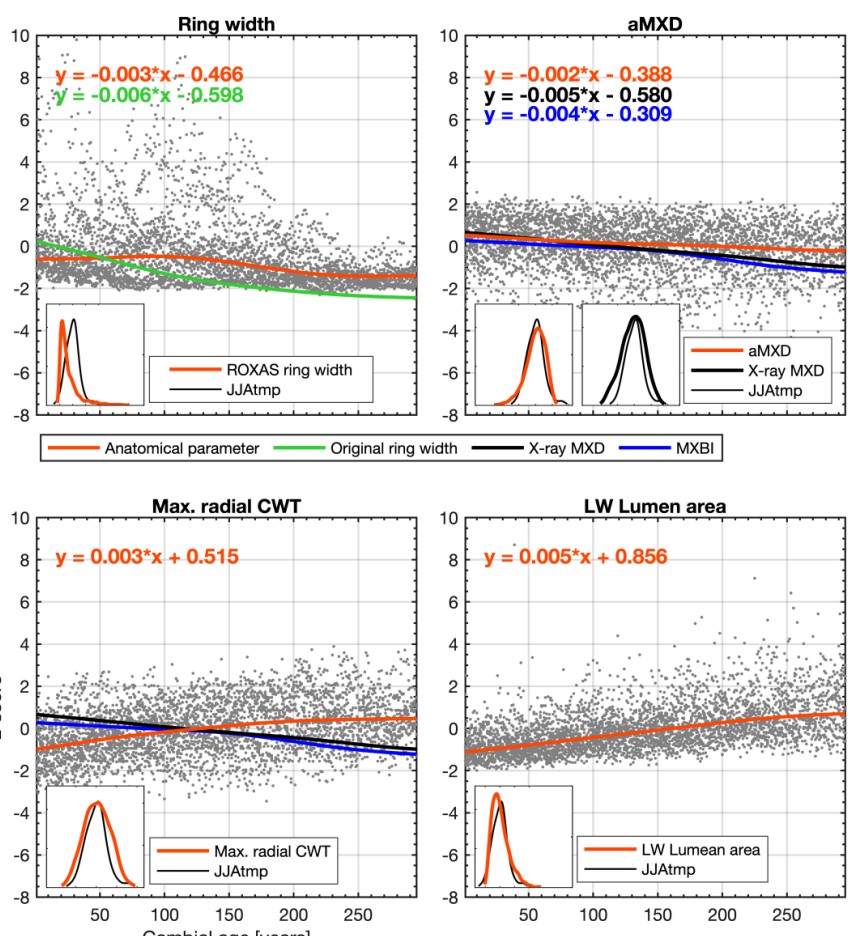

**Figure 9:** *Regional curves of selected anatomical datasets. The scatterplots represent individual, non-detrended and cambial-age aligned (first year in each series), anatomical measurements. The orange curves are the medians of the individual data points smoothed with a cubic smoothing spline. Note that no pith offset adjustments have been made on the time series. For comparison, we also add smoothed regional curves of the MXBI and X-ray derived MXD datasets. Linear regression equations for each RC are included in the plots. Probability density functions of the parameters are provided at the bottom of each panel, together with warm-season temperatures for comparison of distribution and prevalence of outliers in the proxy and in the temperature target.*



**Concluding remarks**

Tree-ring based reconstructions of pre-industrial climate provide a key insight into Earth's present and future changing climate, yet their full potential will remain unexploited without a concerted effort to overcome several critical challenges. This study is part of a larger ongoing synergetic effort (e.g., Björklund et al. 2020, and other work currently in preparation) directed at exploring the efficacy of highly temperature sensitive tree-ring data frequently used in large scale temperature reconstructions (e.g., Wilson et al. 2016), with the ambition to improve upon these existing records using dendroanatomical techniques. This is because dendroanatomy represents the direct morphological refinement of current state-of-the-art microdensitometric techniques where it is possible to have within-ring specific location of the measurements down to the cellular level (von Arx and Carrer 2014).

In summary, based on the collective comparison between the new wood anatomical dataset of *P. engelmannii* from the Columbia Icefields and the two predecessors X-ray MXD and MXBI, we are able to draw the following conclusions:

1. Maximum radial cell wall thickness and anatomical MXD are the two most promising wood anatomical proxy parameters for estimating past temperatures, each explaining >45% in instrumental, high-pass filtered, July-August maximum temperatures. Both parameters display a comparable climatic imprint and strength of signal to the current-state-of-the-art X-ray derived MXD. It does, however, appear that the stability of the temperature signal over time is more robust for the maximum radial cell wall thickness than for X-ray MXD.

2. For these anatomical parameters, the number of trees needed to reach the commonly accepted quality threshold for chronologies used in dendroclimatic analyses is, for our experimental site and species, exemplary with just six trees. However, this high common signal strength is matched by the X-ray MXD parameter and thus does not constitute an obvious advantage by itself. Nevertheless, if the temperature signal is more stable in maximum radial cell wall thickness, it is advantageous to know that very few trees are needed to reach chronology confidence. This is especially true given that the problem of fading records, i.e. the general decrease in sample replication and between tree correlations back in time (Esper and Büntgen 2021), poses a severe constraint to almost all chronologies extending up to or beyond the last millennium.

3. The higher resolution of dendroanatomy appears to positively influence the high-frequency temperature signal stability. Using anatomical parameters as opposed to density parameters, be it from X-ray or anatomy, may also be beneficial for data quality and the mechanistic interpretation of the proxy record. However, further research is needed to consolidate this and other important potential effects regarding the low



frequency fidelity of long-term temperature reconstructions based on X-ray
densitometry.
Finally, despite the encouraging results detailed herein, it is necessary to continue to extend
this dataset by adding more series from multiple age classes and across the last millennium
to more thoroughly evaluate the multi-centennial to millennial scale variations of this key
temperature proxy site. The work detailed here is the first piece of a puzzle to explore
dendroanatomy of the *P. engelmannii* sample set for the Columbia Icefield area in Canada,
formerly analyzed with X-ray and BI techniques (Luckman and Wilson 2005). As such, it also
represents the longest (1585 – 2014 CE) dendroanatomical dataset currently developed for
North America.

**Author contributions**

KS and JB conceptualized the research and obtained the funding to support it. MF performed
the dendroanatomical measurements, using wood material collected by BL and RW. GvA
aided the interpretation of the dendroanatomical data, and MR of the BI-measurements. KS
carried out the analysis and drafted the paper. All authors contributed to the planning and
structuring of the paper.

**Data availability**

The dendroanatomical chronologies from the Icefields area, Canada, will be available on
request.

**Competing interests**

The authors declare that they have no conflict of interest.

**Acknowledgments**

This work was financed by FORMAS (Grant No. 2019-01482 to KS), the Swiss National
Science Foundation (Project XELLCLIM no. 200021_182398 to GvA.) RW received funds
through the US National Science Foundation (NSF) Grant AGS 1502150 for the MXBI
measurements. MR was supported by the Czech Science Foundation project REPLICATE
(20-22351Y).

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
