# Peer review of "Prospects for dendroanatomy in paleoclimatology – a case study on *Picea"

_Climate of the Past, 2021_

## Author Comment (AC1)

**Reply on RC1**

This manuscript investigates the potential of dendroanatomical features of high-elevation Engelmann spruce from the Canadian Rockies as paleoclimate proxies. In doing so, the authors develop the longest dendroanatomical dataset for North America. I feel this study very interesting as it not only provides a comprehensive assessment of a relatively new tree-ring proxy type, dendroanatomy, but also attempts to compare dendroanatomical parameters with those obtained using the existing techniques - X-ray densitometry and blue intensity. I also appreciate that the authors test for the resolution related biases in BI and X-ray density and use two climate datasets to avoid biased results. This study finds that maximum radial cell wall thickness and anatomical MXD are the two most robust proxies of summer temperature and may be superior to MXBI and even X-ray MXD, where the latter has long been recognized as the most temperature-sensitive tree-ring proxy. These promising results will encourage dendroclimatologists to use anatomical parameters to better understand regional and large-scale climate history spanning centuries to millennia.

Overall, this study is valuable, although there are a couple of weak parts, that are outlined below.

*We thank the referee for acknowledging the importance of our work, and also for the time and effort that she/he have put into thoroughly assessing our manuscript. We have taken the opportunity to revise our work following this helpful suggestions. Provided below is a detailed description of how we have incorporated the requests and suggestions (shown in blue italics). We also provide a version of the manuscript where all changes are tracked.*

**Major comments:**
My primary comments are about the long-term trend analysis in section 3.3. In dendroclimatology, it is widely accepted that many tree-ring parameters, even including dendroanatomical ones (see fig. S7 of Björklund et al., 2020), exhibit age effects which may mask true low-frequency climate variability embedded in tree-ring data. Thus, detrending is fundamental to remove these non-climatic, long-term trends, while it may also be challenged in specific cases. Indeed, it may be difficult to disentangle the longterm climate and age signals in even-aged samples as is the case of this study. Here, the authors try to interpret long-term trends of different parameters based on non-detrended series. Though "a robust picture of long-term trends needs RCS-type detrending" is acknowledged on Lines 578–579, I still wonder whether the analysis of the long-term trend based on the non-detrended data (mixed with both age-related and climatic trends) is appropriate. Moreover, the use of instrumental temperatures here to compare with nondetrended tree-ring data is confusing as non-detrended tree-ring data contain more information than climate.

While the authors' analysis on the "high-pass" time series is good enough (not necessary to change), I would suggest using the detrended dataset for discussing the long-term trends (Lines 578–600), unless reasons and meanings of not using detrending could be sufficiently justified in the manuscript. The signal-free approach (Melvin and Briffa 2008) may be efficient to preserve more low-frequency climate signals. If the signal-free regional curve standardization does not lead to success due to low replication, signal-free approach plus age-dependent spline would likely be useful (see Wilson et al., 2019, Heeter et al., 2020, Wang et al., 2020). I also think even if these detrending methods may result in problematic long-term trends, it is still useful to present/discuss these chronologies rather than the non-detrended series. At least this will show that some tree-ring parameters are likely facing issues of retaining low-frequency signals, and further studies are thus needed. It might be useful to plot the age-related trend of each tree series for each parameter similar to (if necessary, replace) Figure 9, when discussing the success or failure of using these detrending methods. In addition, the reconstruction of Luckman and Wilson (2004) might be used as a reference to assess the long-term trends of detrended chronologies.

*We thank the referee for this important remark. We agree that dendroanatomical data may too be subjected to age-related trends (as shown by Björklund et al., 2020) and also suggested by the results of this study (see e.g., fig. S4 where aMXD and Max. radial CWT clearly reveal contrasting trends). With that said, we do not agree that the use signal-free approach is appropriate in this context. Our study is based entirely on living trees, which means that they very likely share a climate signal at lower frequencies even if they are aligned by cambial age (i.e. trend in signal). This means that any attempt, even using signal-free approaches, will provide indices that likely have to be revised when implementing the same technique on a large multi-generational material (ongoing work). By providing such indices the current study would suggest certainty where there is little, leaving the readers misled and confused when the long-term trends will have to be revised in the next ms presenting the full 1000y dataset. We have now added text to sect 2.2 to clarify this point and argue for not using RCS or the signal free approach in this first ms. We have also followed the suggestion made by the referee and removed the comparison between instrumental temperature and nondetrended tree-ring data from fig 6, and also removed sect. 3.3 (previous version of the ms) discussing the long-term trends in the dataset.*

**Additional comments:**
Line 32: Long-term secular trends, please refer to the major comments. *See our previous response.*

Line 49: It is better to specify what are the "important climate periods". *Clarified (we here refer to the MCA and LIA).*

Lines 70–71: Perhaps also add Björklund et al., 2020 (see the recommended references in the end). *Added.*

Line 80: should use "has become"... *corrected.*

Lines 97–102: Try to merge the two sentences. The second sentence is quite similar to the latter half of the first sentence. *Merged.*

Lines 104–105: Avoid saying that 15 trees are well-replicated. A well-replicated dataset may refer to a collection of hundreds of trees. *Sentence rephrased.*

Line 109: X-ray radiography is not a new technique and Dendro2003 is a system designed about 20 years ago. There are now many advancing techniques providing higher resolution density data, such as computed tomography (see Van den Bulcke et al., 2019).
"the state-of-art" here should be removed. *We have followed the reviewers remark and removed the "state-of-the-art" here and elsewhere in the ms. However, we would still like to point out that although the Walesch system is arguably 30-40 years in the making it is still the state-of- the art in terms of data quality, homogeneity and resolution. We acknowledge that there are other more advanced systems around but they are not superior in data quality (see Björklund et al., 2019). Neither have these techniques yet generated sufficient amount of data for them to be tested and evaluated on a broader scale.*

Line 114: Be careful, en dash (–) should be used to represent the range in all cases, rather hyphen (-). Please check throughout the manuscript. *Checked and corrected.*

Line 115: What is the meaning of "pivotal locations" here? I think every location lacking a good proxy record can be pivotal for paleoclimatology. Consider removing "pivotal". *Removed.*

Line 133: Try to move the arrow a little to indicate the correct location. A bit misleading here. *The arrow is now removed from fig 1, and the caption is re-worded.*

Line 136: Athabasca Glaciers is not indicated in Figure 1A). Please try to indicate it on the map, or remove the "Athabasca Glaciers" in the figure caption. *Removed.*

Line 145: It is better to use "immersed" instead of "washed".
*Changed to "refluxed". We used a Soxlet setup, where alcohol was circulating. "Immersed" is therefore in our opinion not the right word to describe the method.*

Line 158: The averaged cell wall thickness is not discussed in the manuscript. It is not necessary to keep it here. CWT should be moved to Line 157. *Average cell wall thickness is not discussed in the ms, yet it is used to derive other parameters. For every cell the average CWT is computed. To get aMXD the ratio between average CWT and total cell area is computed. For each 20 micrometer wide band, the 75$^{th}$ percentile of these values are selected to get a profile across the ring. The maximum value of the profile is then selected to get the aMXD value of that year.*

Lines 159–161: What is the tangential width of the measurement window? The 75th percentile of what values? Are only the 75th percentile values used or 0–75 percentile used? Please specify. In addition, by delimiting the 20 um bands some tracheids would be separated into different bands, I wonder how the CWT was obtained? It seems hard to measure "two radial and two tangential cell walls per tracheid cell" if tracheids are separated. Perhaps more details are needed so that the experiment could be repeated. *This section has been expanded to clarify the methodological questions raised by the referee.*

Line 179: Should indicate the version of CooRecorder. *Info added – ver. 8.1. We note however that the program version do not matter in this context, since the MXBI measurements are not affected by the program version.*

Line 197: Figure S3 is not mentioned anywhere in the manuscript. Perhaps need to refer to Figure S3 here and re-order the supplementary figures. *Done.*

Lines 204–205: The justification for not using RCS is not convincing enough. In my view, RCS works efficiently in cases where dead trees are also included, as it could avoid the "segment-length-curse". If RCS does not work for the 15 living trees, the low sample replication and un-uniform growth patterns are more likely the reason. So, consider clarifying here that low replication hampers the use of RCS. I also suggest giving some supplementary graphics of RCS chronologies to explain why RCS is not suitable. In line with my major comments, it is worth trying signal-free RCS and signal-free age dependent spline smoothing as well. *See our previous response (under "major comments").*

Lines 213–215: Is there a particular reason why the robust mean is not used here? Specify if possible. However, it is not a big problem. It is better to use "detrended data" (or "Spline smoothed data", if a second detrending method is used; see major comments) in the manuscript because some medium-frequency signals may still be retained by using the 35-yr spline smoothing. "High-pass filtered data" sounds like all low frequencies are filtered out. *Done, changed from "high-pass filtered" to "detrended data" according to suggestion.*

Lines 220–221: Please avoid writing "sufficient sample depth" here. The minimum sample depth is only 9, which even doesn't lead to an EPS>0.85 for many parameters according to Table 1. *Removed and rephrased.*

Line 229: Caution, the term "cross-correlation" is wrongly used in this manuscript. Generally, in statistics, the term "cross-correlation" represents the lagged correlations between time series, rather than correlations across time series. "Pairwise correlation"

should be used in the entire manuscript.
*We appreciate this clarification, we have now changed to "pairwise correlation", both here and throughout the rest of the manuscript.*

Line 241: The correct citation should be "St. George and Luckman (2001)". Change it also in the reference list. *Corrected.*

Line 245: Citation or URL of Meteorological Service of Canada should be added. In addition, I couldn't find the gridded data spanning 1895–present from the Meteorological Service of Canada. Where the data could be accessed? *Citation added.*

Line 264: What is the time period used to calculate the r bar and EPS? The period 1585–2014 here is not consistent with Line 219 which describes 1700-1994 is used for the subsequent analysis. Please be consistent. *This is now clarified both in the results section and in the table 1 caption. All the statistics in table 1 have been computed over the common 1700–1994 period.*

Line 266: If there is only one study cited here, the sentence should be: "with a previous study..." *Corrected.*

Line 272 and Table 1: How the n for EPS = 0.85 (the last column of Table 1) is estimated? Some n is even greater than the actual number of samples. I guess "n for EPS = 0.85" is calculated based on the rbar and the equation of EPS. Please clarify this somewhere perhaps with an equation, e.g., at the bottom of Table 1. *Yes, the estimation is based on the rbar statistics for each tree-ring parameter and the equation of the EPS. This is now clarified in the table caption.*

Lines 301 and 414: See the comment for Line 229. *Corrected.*

Lines 336–337: Should use "The first two components together represent 68.1% of the total variation". Should also clarify what correlation is used, Pearson's r? *rephrased and clarified.*

Lines 339, 390, and 418: see the comment for Lines 213–215. *Changed to "detrended data", here and throughout the manuscript.*

Lines 364–366: Perhaps also useful to compare the seasonal responsive window with Picea species in the North American continent. For example, black spruce in the eastern North American boreal forest, in similar latitudes. *We thank the reviewer for this suggestion. We have now added a short comparison of our results with the recent (2020) study by Wang et al. on black spruce from eastern Canada.*

Line 391: "Monthly or seasonally averaged temperature". *sentence re-phrased (also taking into account the suggestion made by referee 2)*

Lines 411–413. Besides the low measurement resolution, the color sensitivity of the BI method may also affect the signal strength of MXBI to some extent. It is hard to ensure that all the measured wood cores are completely free of color biases. It is thus highly appreciated to use a few sentences somewhere to illustrate that weaker signal strength may also result from the color sensitivity of MXBI even for unstained living trees, see Wang et al., 2020. This is a fair discussion about the BI method. *We understand the concern raised by the reviewer, and the MXBI discoloration issue is already mentioned in our work (lines 613-622). Sample discoloration, however, has a gradual effect on the MXBI chronology, often manifesting as a long-term trend. Here, the data has been detrended with a 35-y spline, which should remove the effect of sample discoloration.*

Line 430: consider referring to Figure S3 after "in the widest rings". *Done*

Line 439: r² should be used instead of r2. Sometimes R2 is also used in the manuscript. Please keep the expression consistent. *Changed.*

Lines 485–487: Consider simplifying the sentence: "were further assessed by a split period calibration procedure (1901–1948 and 1949–1994)". *Changed accordingly.*

Lines 488–489: Refer to the major comments. *See our previous reply. Figure 5 has been modified to display the results only for the detrended data.*

Lines 513: Since the r2 distribution consists of 1000 values, it is better and more logical to perform some statistical test (e.g., two-sided student's t) to show whether correlations of different parameters are similar or not. *We thank the referee for this suggestion. We have now performed a two-sample t-test to whether there is a significant difference in calibration statistics between parameters.*

Line 516: Figures 7–9 were mentioned only once each in the main manuscript. Maybe should consider moving them to the supplementary material. In addition, "correlated width" should be changed to "correlated with". *Figure 9 moved to the supplement according to suggestion. Figures 7 and 8 are however kept in the ms as these are important for the discussion.*

Line 530, figure 6: the top-left annotations appear not complete. Only Max. radial CWT in A) and only aMXD in B)? Since aMXD and X-ray MXD are more directly comparable, why they are plotted in two separate plots? Perhaps show chronologies using either signal-free RCS or signal-free age-dependent spline here, after the major comments are taken into account. *The plots in fig 6 (now fig. 5) are now changed to display detrended data (standardized using a 35-year spline). We have also removed all information from the plots based on non-detrended mean chronologies. aMXD and x-ray MXD are now shown in the same panel.*

Lines 550–611: please refer to the major comments. *See our previous response concerning the detrending issue.*

Line 560: Should use "lower frequencies". *Changed*

Line 625, Figure 9: Why JJA temperature is used here? Ring width contains JJ signal and density parameters contain JA signal. *Figure moved to supplement. JJA is used here as it encompasses the signal of both RW and the density parameters.*

Line 655: Please remove "the state-of-art". *Removed. However, see our previous comment.*

The reference list: check the format on Lines 729–731; Line 745: "IAWA J"; Line 755: "Science"; Line 795: remove "Verona"; check Lines 804–806; Are Lines 829, 832, and 870 necessary? Line 864: volume and page numbers should be added. *We thank the reviewer for taking time and checking the reference list. The format issues are now corrected (except for Ln 864, where volume and page no are not available).*

References cited in this review:
Björklund et al., 2020. Dendroclimatic potential of dendroanatomy in temperaturesensitive Pinus sylvestris.

Heeter et al., 2020. Late summer temperature variability for the Southern Rocky Mountains (USA) since 1735 CE: applying blue light intensity to low-latitude Picea engelmannii Parry ex Engelm

Luckman and Wilson, 2004. Summer temperatures in the Canadian Rockies during the last millennium: a revised record.

Melvin and Briffa, 2008. A "signal-free" approach to dendroclimatic standardization.

Van den Bulcke et al., 2019. Advanced X-ray CT scanning can boost tree ring research for earth system sciences.

Wang et al., 2020. Temperature sensitivity of blue intensity, maximum latewood density, and ring width data of living black spruce trees in the eastern Canadian taiga.

Wilson et al., 2019. Improved dendroclimatic calibration using blue intensity in the southern Yukon.

*Thanks for this clarification.*

---

## Author Comment (AC2)

**Reply on RC2**

The paper „Prospects for dendroanatomy in paleoclimatology – a case study on Picea engelmannii from the Canadian Rockies" by K. Seftigen et al., explores the paleoclimatic potential of a broad set of dendro anatomical proxies, in particular focusing on the relationships between "new" proxies (mostly related to wood anatomy) and proxies that are relatively more known, such as X-ray maximum latewood density (MXD) and blue intensity (MXBI), and already used in the boreal environments of North America. My general impression about this manuscript is very positive, the paper deals with a timely issue, and such an assessment of the strengths and weaknesses of dendro anatomical proxies is a much-needed help for anyone working with "alternative" tree-ring paleoclimatic proxies. It is also notable that long chronologies of wood anatomical parameters are finally emerging even for the North American continent. For this reason, I can easily foresee that this work has the potential to become a highly cited reference in the field.

That said, I think there are still a few points that require attention.

*We are glad that referee #2 found our research worthy of publication and we appreciate the opportunity to revise our manuscripts following the helpful suggestions. Below we provide further detailed description of how we have incorporated the requests and comments. Reviewer comments are followed by our replies in blue.*

**Major issues**

The first major issue is related to the chronology development of wood anatomical parameters: in lines 159-165 it is specified that wood anatomical parameters were calculated as the 75th percentile within 20 µm wide bands parallel to the ring boundary. Nothing to say against this tree-ring partitioning approach, but it is not clear how the data obtained for each band were treated: usually, when the tree ring is partitioned, then multiple chronologies are developed. Since I do not see this in the manuscript, I assume that these data were somehow averaged. If so, it should be specified. In any case, the procedure of chronology development should be clearer, avoiding pointing to third papers.

*This may be a misconception. We move the 20micron band radially over the tree-ring and for all cells enveloped by the band we take the 75percentile. When the band has been moved across the full ring, we obtain a profile of values. We take the highest value in the resulting profile for all the maximum parameters, the lowest value of the resulting profile for the minimum parameters and the average of all values within the latewood for all latewood parameters etc.. We have now clarified this methodological aspect in the manuscript (see also related comment by referee #1).*

Secondly, I have major concerns related to the procedure for testing temporal stability of dendroclimatic relationships in dendro anatomical proxies (Ln250 - Ln259 & Ln452 - Ln 473):

1) if the goal is to test the temporal stability of dendroclimatic relationships obtained for the monthly data, why shift from using monthly data to daily data, using a different climatic dataset (the Berkeley Earth dataset) that is, in addition, experimental and that You seem not to trust completely (see Ln. 461-464)? I think this is simply wrong and cannot be accepted unless a thorough comparison between the two datasets (CRU TS v4.03 and Berkeley Earth dataset) is performed to check if there exist substantial differences between the two.

*We thank the reviewer for suggesting this complementary test. We have now evaluated the similarity between the CRU and Berkley products by aggregating the daily Berkley product into monthly resolution (by averaging the daily values for each month) and correlating it against the tree-ring parameters (the same way as we have performed the correlation analysis between the CRU temperatures and tree-ring parameters in fig 3). These new results, added to the supplement (Fig. S6), show identical patterns to the CRU-based tree growth-climate*

*correlations (fig. 3), which makes us draw the conclusion that the two climate products are indeed comparable and can be used in a experimental setup as the one that we use in our work.*

2) if daily data are going to be used (and I am a huge supporter of daily data, whenever available, especially with wood anatomical data) why use them in 30-days fixed windows (so basically reconducting to a monthly analysis)? This is not the approach proposed by Jevsenak and Levanic (2018), which I would actually recommend following, testing moving correlations over the same 30-year period but using temporal windows starting from i.e. 14 days up to whatever length You'd like to test (30, 60, 180 days). That would help better target the temporal windows of climatic sensitivity described in Figure 3 and Figure 5.

*Actually, the R-script developed by Jevsenak and Levanic (2018) both offers a fixed (similar to our approach) and variable (as pointed out by the referee) window width when calculating the correlations between daily met data and tree-ring chronologies.*
*We would like to keep the analysis at it stands because the 60, 180 days widows are redundant since we already check the smaller window size of 30 days. The similarity of the correlation in one pixel to the next means that the window of temperature signal can be extended one day at the time. If the correlation coefficient is stable 30 days in a row, it means that the signal extends for two whole 30 day windows -> 60 days and so forth.. We refrained from doing the analysis with 14-day windows because the analysis would be sensitive to spurious correlations and difficult to interpret. Moreover, since we usually have significant monthly correlations with two months (Figure 3), we can detail the results insofar that if we find strong correlations for 15 pixels in a row and thereafter a drop, it means that the signal window length is 45 days starting on the Julian day of X.*

A third major concern is related to the overall presentation of the paper. Despite being well written and accompanied with high-quality pictures, the manuscript is, in my personal opinion, exceptionally long, and at some point, allow me to use this term, becomes exhausting to read. I refer, for example, to the analysis related to the biases in the MXBI technique that are introduced in lines 190-199 and then discussed in lines 411-450. Honestly, I do not see the point of this long analysis and discussion. I am not criticizing its soundness nor questioning its interest. But to me, it looks like material for a stand-along paper, a deep technicality related to the blue intensity with weak (or at least not immediate) implications for dendro anatomy within the context of this manuscript. I believe that the strength of this work is that it is exploring important technical details related to those dendro-anatomical proxies that are less known in this region. It should not focus on going into deep technical details related to proxies or techniques that are already relatively more known and applied.
Another example is in lines 368-385. This long paragraph discussing the climatic response of EW features could be easily avoided, since it is evident throughout the entire manuscript, and according to Table 1, that EW signal is not predominant at this location. Since the paper is already very long, this long (and sometimes repetitive) discussion on EW density and lumen area could be avoided. The last example could be section 3.1, where several times concepts discussed in the following sections are already mentioned. In general, I suggest careful evaluation throughout the manuscript if certain paragraphs and/or concepts could be removed or rephrased to improve the readability of this paper.

*We have followed the advice and shortened the manuscript where we find appropriate. For example, we have removed the paragraph 368-385 according to the suggestion, and also removed the last section discussing long-term trends (following the suggestion made by reviewer 1). We have also split the results/discussion section into sub-sections including new subheadings "Temporal signal stability" and "Possible implications of measurement resolution on climate signal". We hope that this will make the text more "airy" and easy to navigate.*

*We have decided to keep the discussion around MXBI and its potential resolution bias as these results are significant when interpreting the strength and seasonality of the temperature imprint withing the various parameters. Moreover, the analysis around the MXBI resolution issue was appreciated by referee #2, which indicates that these results may be of interests to the community and worthy a publication.*

**Minor issues:**

The paper introduces the longest dataset of dendro-anatomical parameters for North America but there is not a figure showing such 1585-2014 chronology in the paper for any parameter. I suggest adding it, even if the analysis is focused on the 1700-1994 period. Moreover: why is the chronology truncated in 1994? And the data in Table 1 refers to the 1585-2014 or 1700-1994 timeframe? Please clarify. *We have followed these suggestion and 1) added a new figure to the supplement showing the full 1585-2014 chronology for the aMXD and Max. radial CWT (fig. S4), clarified in the ms the reason for the truncation to year 1994 (to match the X-ray dataset), and also 3) provided info on which period the table 1 statistics are computed (the common 1700-1994 period, info added to the table caption).*

Lines 320-331 Could be integrated into the next section. *Integrated and also partly removed to shorten the text.*

Lines 360-361 "Even though the parameters describe two temporally distinct temperature signals, both are encapsulated within the short June-July-August period". What are the two temporally distinct temperature signals? Please clarify. *Sentence removed.*

Lines 363-364 "…average monthly temperatures rise above 0 °C only in four months of the year". In Fig 1c it is actually five months. *Typo, now changed to five months.*

Lines 364-366 "This window is substantially shorter than the single but wide target season observed in the latewood anatomical traits of P. sylvestris growing in temperature-limited environments in northern Scandinavia". I don't understand this sentence (i.e. what is the "target season observed in latewood anatomical traits"?), please be more clear. *Rephrased to "[…] comparing our results with the previous study of (Björklund et al. 2020) on latewood anatomical traits of P. sylvestris in northern Scandinavia, where the temperature response window extends from April to September." .*

Lines 368-385 Regardless of my previous comment (see above), I would like to add a few words about this. The reversing relationship observed between June-July temperature and EW density and lumen area might be due to an effect of precipitation, or in general of moisture availability, at that time. The initial stages of wood formation (hence, the anatomical features of the cells formed firstly along the ring) are highly dependent on water availability, which determines cell turgor in the enlargement phase. This is generally reflected in the EW lumen features, such as lumen area or lumen radial diameter. As a general remark, I would not say that EW features (i.e. lumen area) are not suitable climate proxies (or at least I would properly contextualize it): this might (is) true for this location and for temperature limited environments, but in arid environments earlywood (not latewood) features (in particular lumen size) are a crucial climatic proxy (not cell wall thickness). *The paragraph the referee refer to is now removed. Nevertheless, we agree with the reviewer that moisture availability in many cases in an important driver for cell enlargement in the EW, especially in drought-prone environments. However, correlation analysis with precipitation for our site and species do not show any significant link to moisture availability, which makes us believe that this environmental constraint is of secondary importance.*

Figure 4 The r coefficients listed on the right side, how are the aggregate months' correlation computed? Is it an average of the two r? Why some values are listed for both

single and aggregated months, and in other cases not, even though the correlations seem equally strong? *We believe that the reviewer refer to fig 3 here? The temperature data have first been averaged over the specific season, and then correlated against the tree-ring chronology to obtain the correlation coefficients. The seasons/monthly windows are selected based on the strongest correlation between the tree-ring data and temperatures. We have now harmonized the figure to make it less confusing: for ring width and all EW parameters we now show June correlation coefficients, and for all LW parameters July-August and August temperature correlations.*

Lines 458-459 Please clarify the meaning of "peripheral ends" and of "elusive". *Re-worded, now reads "The temperature associations at the margins of the target season are, however, more unstable.".*

Lines 484-489 This is a key methodological aspect that lead to one of the main findings of this paper, hence I was surprised to find it here and not in the Methods section. Please move it to the Methods section. Done according to suggestion. *The paragraph has been moved to sect. 2.3 in the methods.*

---

## Author Response (AR2)

We would like to thank the Editor and the two referees for the positive evaluation of our manuscript, and the Climate of the Past for publishing our work. We provide here a version of the article where the minor comments provided by referee 1 have been taken into account.

On behalf of the authors,
Kristina Seftigen